# Rainwater Harvesting as Sustainable Solution to Cope with Drinking Water Scarcity and Urban Flooding: A Case Study of Public Institutions in Lahore, Pakistan

Muhammad Waseem [1],*, Syed Mutahir Ullah Ghazi [1], Nameer Ahmed [1], Muhammad Ayaan [1] and Megersa Kebede Leta [2],*

1   Department of Civil Engineering, Ghulam Ishaq Khan Institute of Engineering Sciences and Technology, Topi 23640, Pakistan; mutahir.ullah1@gmail.com (S.M.U.G.); nameerahmad8@gmail.com (N.A.); ayaaankk@gmail.com (M.A.)
2   Faculty of Agriculture and Environmental Sciences, University of Rostock, 18059 Rostock, Germany
*   Correspondence: muhammad.waseem@giki.edu.pk (M.W.); megersa.kebede@uni-rostock.de (M.K.L.)

**Abstract:** Pakistan is currently facing physical and economic water scarcity issues, which have been further complicated by the rapid increase in its population and climate change. In affected areas, many methods are being used to tackle this problem, among which rainwater harvesting (RWH) provides the best alternative source of domestic water supply. In rainwater harvesting, a mechanism is designed to effectively collect surface runoff during rainfall events from residential rooftops. It has also been found that rainwater has great potential as a source of water supply in residential areas of major cities, such as Lahore, which is the focus of our study. This research paper examines rainwater harvesting as a sustainable solution to address the challenges of drinking water scarcity and urban flooding. The study discusses the benefits of rainwater harvesting, including reducing reliance on municipal water sources, improving water quality, and mitigating the impact of urban flooding. Additionally, the paper explores the use of filtered water points in conjunction with rainwater harvesting systems to provide clean drinking water to communities. The research draws on case studies from various regions to illustrate the effectiveness of rainwater harvesting as a sustainable solution to water scarcity and urban flooding. Ultimately, the study concludes that rainwater harvesting, when coupled with filtered water points, can offer an effective and sustainable solution to address drinking water scarcity and urban flooding.

**Keywords:** rainwater harvesting; water scarcity; urban flood mitigation; water conservation; water storage; groundwater recharge; public water policy; Pakistan

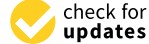



## 1. Introduction

Glaciers and reservoirs worldwide are melting or drying up, resulting in water scarcity in urban areas. This is a significant global issue since it impacts both public and global water supplies [1,2]. Careful management of water is crucial to ensure the survival of the species, improve living standards, promote economic growth, and eradicate poverty [3,4]. The 2030 SDGs, unveiled in 2015, include SDG 6, which aims to "ensure availability and sustainable management of water and sanitation for all" [5]. Water scarcity has become a significant global issue, particularly in densely populated urban areas [6]. Increasing populations, rapid urbanization, and rural-to-urban migration pose significant threats to cities, especially in developing countries [7]. Pakistan has experienced rapid climate change in recent years, which has exacerbated the country's vulnerability to natural resource depletion. Water scarcity is a critical issue in Pakistan's urban areas [8]. The gap between water supply and demand has rapidly widened in Pakistan, and based on current population growth rates, a 31% water shortage is anticipated by 2025. This shortage is expected to worsen until 2050 [9]. Over the past few decades, Pakistan's domestic, industrial, and agricultural

sectors have increased their abstraction of groundwater to meet the growing demands of the country's expanding population. Groundwater is widely available, easily accessible, and considered a reliable source of water [10,11].

Lahore, the largest urban district in Punjab and the state capital, relies solely on groundwater as its source of water, and its depletion rate is increasing over time. The proliferation of private housing societies and industrial activities has led to an increase in groundwater pumping for domestic and industrial purposes, while the shift from agriculture to urbanization has decreased groundwater recharge, putting enormous stress on the groundwater reservoir beneath the city [12,13]. The size of Lahore has nearly doubled in the past 12 to 15 years in terms of area. Unregulated groundwater abstraction and the drying up of the Ravi River, coupled with a lack of recharge operations and surface water supplies, have contributed to annual reduction rates in the water table ranging from 0.43 to 1.5 m, with an average of 0.76 m. A practical and energy-efficient solution to address this issue is rainwater harvesting (RWH), which involves collecting precipitation to recharge groundwater supplies through recharge wells. This approach reduces stress on the current drainage infrastructure from sewage and storm drainage systems and is particularly effective since every one meter rise in the water table saves 0.4 Kwh of power [14]. Given the challenges of diminishing water quality, the ageing water infrastructure, the growing population, and increasing urbanization, it is imperative that we explore sustainable alternatives to our current water distribution system. Rainwater harvesting (RWH) is one of the most frequently considered potential solutions to address these issues. Employing more frequent precipitation collection through rainwater harvesting (RWH) can help avoid many current and future problems related to water. RWH is versatile enough to be used for various purposes, such as washing, cooking, drinking, and irrigation of crops. It also provides an alternative to low-quality water sources, such as polluted surface water during the rainy season, and can be particularly useful during dry spells. Maximizing the use of sparse rainfall is crucial in the current environment, especially in dry and semi-arid regions [15]. Despite all this, rainwater harvesting (RWH) has many challenges and limitations. According to studies, there are many issues related to water quality, excessive costs, RWH management, inadequate volumes of annual runoff, and poor installation and maintenance of RWH systems in many developing countries [16]. In locations with little rainfall and low water supplies, RWH is a practical solution to the problem of water shortages. However, in order to fully realize RWH's potential, a number of issues and restrictions must be resolved: Water quality: Due to the presence of impurities, including pollutants and pathogens, the quality of gathered rainwater may not always be adequate for certain uses. If the water is not properly treated, this might result in environmental and health risks; Excessive cost: Adoption of an RWH system may be significantly hampered by the setup costs. The cost of equipment, installation, and maintenance may be prohibitive for certain people and towns, discouraging them from investing in RWH systems; RWH management: Maximizing the advantages of RWH systems requires effective management. Nevertheless, maintaining an RWH system can be difficult, especially for individuals with little technical expertise and few resources. The system has to be maintained and repaired, storage tanks need to be cleaned frequently, and captured rainwater needs to be used properly; Inadequate volume of annual runoff: Rainfall is a key component in RWH systems, and it can vary greatly from year to year. The amount of yearly runoff may not be sufficient to fulfill the community's water demands in places with minimal rainfall; Poor installation and maintenance of RWH systems: RWH systems' efficiency and efficacy may be compromised by improper installation and maintenance. Leakage, spillage, and pollution of collected rainwater can result from improper installation. To keep the system running well and to avoid system failure, routine maintenance is crucial. RWH systems can help overcome these challenges and promote the sustainable use of rainwater resources.

In many regions of the world, RWH is frequently utilized to address urban flooding and water scarcity. It originated in the Neolithic era (roughly 10,000 BCE to 4500 BCE) [17,18]. Rainwater harvesting (RWH) is a method of collecting and storing rainwater that can

provide a dependable source of water for various purposes, such as household, commercial, and agricultural uses [19–24]. RWH can meet a significant portion of the water demand in underdeveloped nations where household usage is low [25,26]. In multiple Asian countries, RWH is especially significant because its cost-effective implementation is supported by the government [27,28]. To provide more drinking water and irrigation water, more than 5.5 million water storage tanks have been built in China since 2001 [29]. A study carried out in Australia found that the average efficiency of RHW was 87% [30,31]. A study on the collection of rainfall from buildings was carried out in Brazil and found that using rainwater in buildings could significantly reduce the need for fresh water [32]. Sazakli et al. [33] conducted a study to evaluate the potential for potable water savings using RWH in residential areas across the 12 governorates of Jordan. Their analysis revealed that, in 2005, 5.6% of the overall water supply for housing was obtained through this technique. In the United States, the efficiency of residential RWH systems was assessed in 23 cities situated in seven different climatic regions. The findings revealed that the size of the cistern and climate trends had influences on the systems' performance. Overall, the results suggested that RWH can be useful for managing stormwater in American communities. The impact of using collected rainwater on health was also investigated. Trace metal levels were found to be within the recommended health limits in domestic areas but not in highly industrialized ones. Epidemiological studies have shown that using filtered collected rainwater does not appear to increase the incidence of digestive illnesses [34–36]. According to a study by Wang et al. [37], modeling approaches can be used to predict the quality of rainwater. However, the lack of funding, particularly in developing nations, may hinder the widespread adoption of RWH. The volume of water, which depends on unreliable precipitation and requires sizable water storage, is the next problem, followed by the failure to connect with other municipal water components, the overall lack of public help, the quality of the water, and the lack of political commitment [38]. The current study also examined accessible, affordable methods for rainwater filtration and provides recommendations.

Recent technological advancements have increased the demand for remote sensing and geographic information system (GIS) techniques as instruments for estimating RWH potential [39]. Remote sensing is a method of gathering data about an object without ever touching it. Applications of remote sensing imagery include mapping of land use and land cover (LULC), forestry, urban planning, and archaeological research [40]. According to Burrough [41], GISs are a collection of technologies used to gather, store, analyze, and present geographic data from the actual world for a specific set of applications. Water resource management has made extensive use of hybrid GIS applications [42–45]. Geospatial technologies, such as GIS and remote sensing (RS), are heavily utilized in water resource planning and management [46]. This is due to their adaptability for employment in various geographic contexts and their ability to facilitate spatial investigations with a wide variety of datasets representing biophysical and anthropogenic aspects [47]. To apply a standard RWH technique in arid and semi-arid regions (ASARs), Ammar et al. [48] devised, compared, and established three sets of criteria for choosing optimal RWH locations and their associated qualities. Mahmoud and Alazba used RS and a decision support system to determine the locations in Egypt that would be best for long-term RWH and storage. Weerasinghe, Schneider, and Loew [49] focused on utilizing remote sensing (RS) and a geographic information system (GIS) to create a comprehensive framework for assessing water management. The model proposes potential locations for preserving soil moisture and storing water on farms. Studies using runoff modeling, GIS, and RS have identified suitable sites for RWH [49–51].

The current research objectives were to assess how large public institutions' rooftop rainwater collection systems could contribute to meeting the demand for potable water and to establish a link between rainwater harvesting and institutional and municipal water management. The current study also identified rainwater storage tanks and recharge wells to decrease flood inundation in Lahore city through surface runoff harvesting. By

providing a primary overview of the RWH in the study area, this study aimed to direct further research to determine the precise amounts of water that can be harvested and create useful management planning. Figure 1 illustrates the trends of rising population and decreasing per capita water availability over the decades, which is a serious concern.

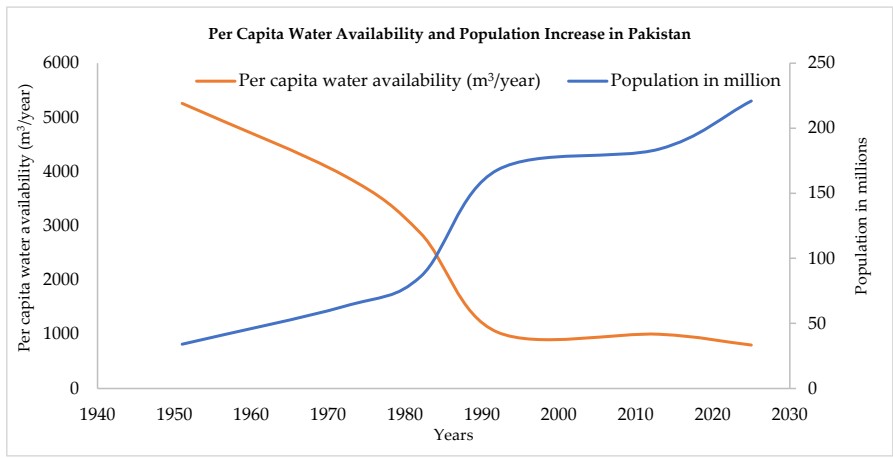

**Figure 1.** Per capita water availability and population increase in Pakistan using data obtained from the Water and Sanitation Agency.

## 2. Materials and Methods

This research proposes a practical approach to mitigate urban floods and replenish natural water resources in Lahore by harnessing rainwater runoff. The study employed a robust methodology that involved analyzing images to classify and digitize institutional rooftops. The volume of rainwater was estimated by calculating the rooftop area, which was then compared to the water demand to determine the potential for rainwater harvesting. In addition, groundwater recharge zones and underground storage tanks were identified. The harvested rainwater was then filtered using various methods for domestic and other applications. The methodological framework adopted in the current research study is shown in Figure 2.

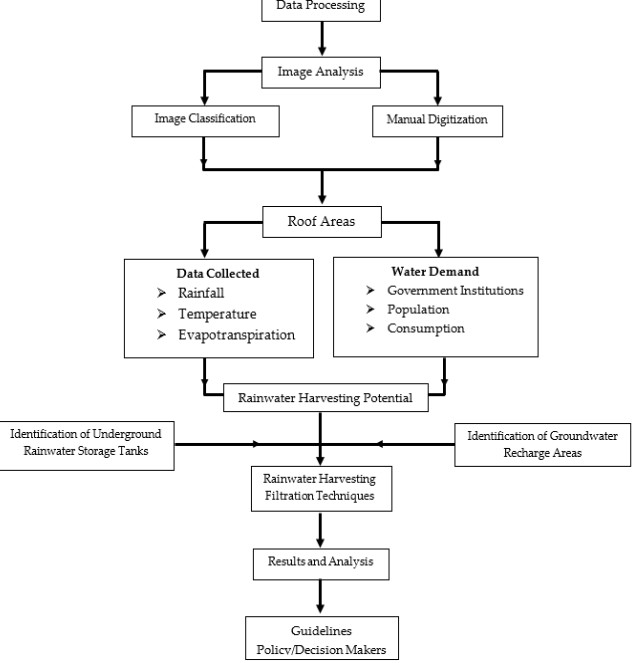

**Figure 2.** Methodological framework of the current study.

### 2.1. Study Area

Punjab's capital, Lahore, is Pakistan's second-largest metropolis and geographically located at 31.54944° N and 74.34333° E. The district covers 1772 km² and, due to its proximity to Sheikhupura in the north, a significant portion of its urban area was previously agricultural land. Lahore's urban area has nearly doubled in size over the last 12 to 15 years. According to the 2023 census, there are nearly 14 million people, with a population density of about 6300 persons/km². The urban population comprises 82.4% of the total. Lahore has industrial areas, including Kot Lakhpat and the new Sundar Industrial Estate. Therefore, the area's increasing urbanization, including the industrial enterprises, is creating numerous significant challenges and opportunities in the new century. Meeting the population's needs for water and sanitation is one of these challenges. In this context, the Water and Sanitation Agency (WASA) of Lahore Development Authority has launched a Master Plan with a 25-year planning horizon and identified projects for the establishment of physical infrastructure for water supply, sewage systems, and wastewater treatment, with comprehensive development likely to cost Lahore district as it aims to meet current and future needs for efficient and effective service delivery. The study area is shown in Figure 3.

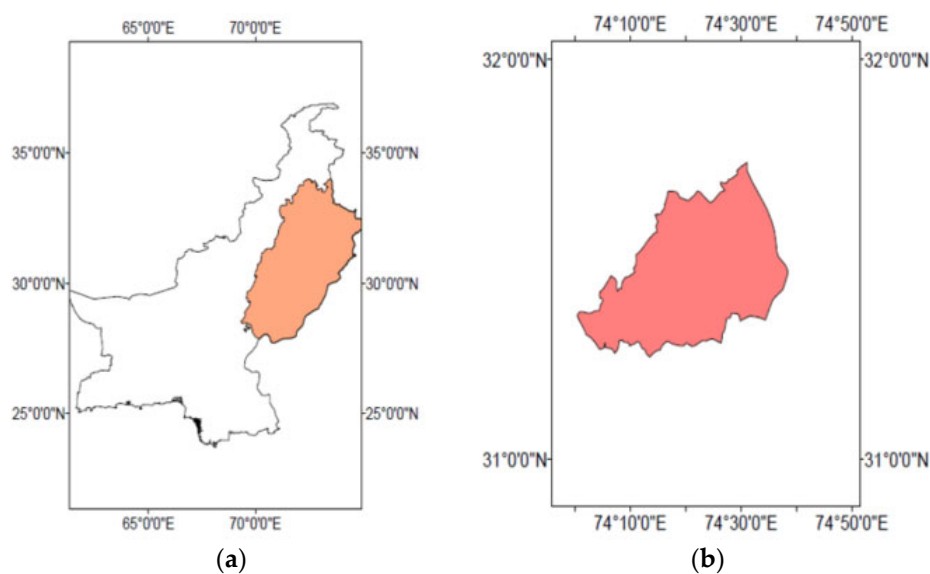

**Figure 3.** (**a**) Location of Punjab province in Pakistan; (**b**) location of Lahore district in Punjab.

### 2.2. Current Situation of Water Sanitation in Lahore

The city of Lahore experiences an average annual rainfall of 629 mm, with the majority (74.7%) occurring during the monsoon season. The current drainage system in Lahore can effectively manage the runoff during the monsoon season. However, rapid urbanization and population growth have led to some parts of the city's drainage system receiving sewage due to an excess in the sewer system. The present drainage system carries combined stormwater and sewage flows from the street's hose connection to the drainage region's drain. Furthermore, extensive construction activities in the area have made it difficult for the drainage system to absorb rainwater runoff directly into the drain. Consequently, rainwater accumulates on rooftops and in streets, parking lots, and other paved areas, leading to overflows that are discharged either directly into the sewers or into nearby parks and other areas, as shown in Figure 4a,b. Stormwater drainage is a significant environmental risk and community issue in Lahore's highly populated neighborhoods, particularly during the monsoon season. Runoff flows along the roadways and accumulates in low-lying areas, creating ponding zones that cannot always be evacuated by surface drainage. In such cases, the sewage system is used to discharge the impounded runoff. During catastrophic events, significant residential and commercial sectors in the city become flooded, putting these

critically ponded regions in the spotlight. The amount of water is too much for the surface drainage system to handle, aggravating the problem.

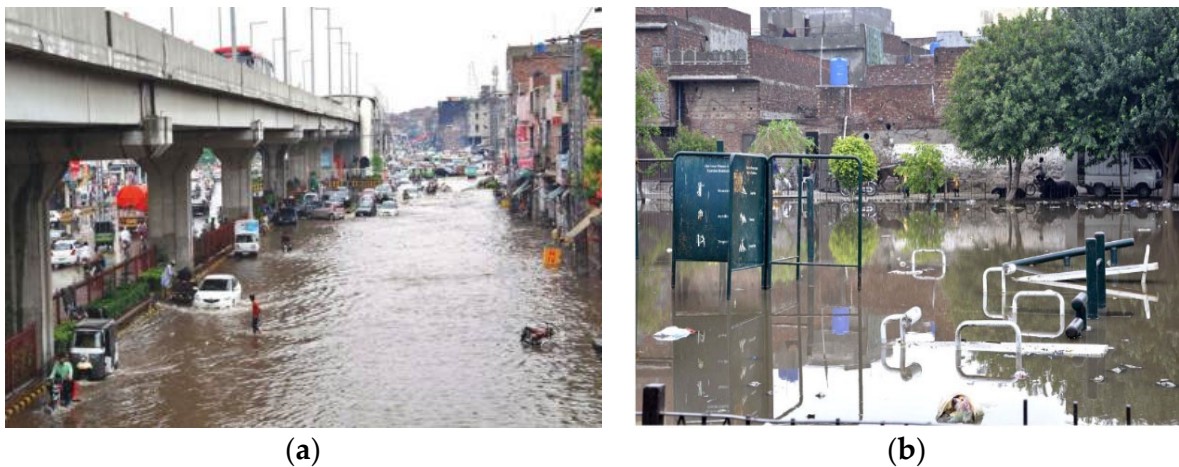

(**a**) (**b**)

**Figure 4.** (**a**) Urban flooding in Lahore during the 2022 monsoon season [52]; (**b**) ponding in a public park [53].

*2.3. Data Collection*

2.3.1. Rainfall Data

There are multiple rain stations scattered throughout Punjab, and only two of them provide data relevant to Lahore (Appendix A). The Thiessen polygon interpolation method was used to divide the catchment area under each climate station. In order to calculate the amount of rainwater that could potentially be harvested, the average monthly rainfall (1980–2020) from the two climate monitoring stations was analyzed and is shown in Figure 5.

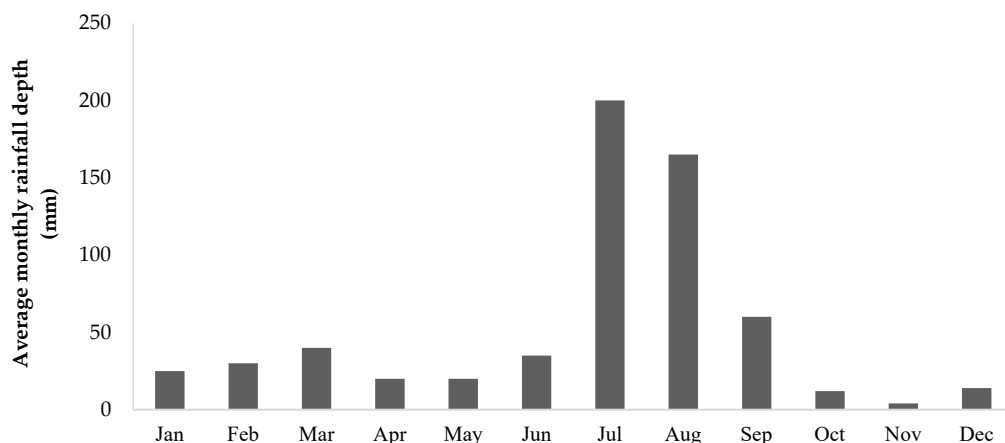

**Figure 5.** Average monthly rainfall in Lahore city based on data provided by Pakistan Meteorological Department for the period 1980–2020.

2.3.2. Construction of Surface Water Natural Drainage with ArcGIS

To identify potential ponding sites and determine the optimal locations for underground water storage tanks, we conducted an analysis of stream networks of various orders using ArcGIS. Our analysis revealed that areas where multiple stream networks converge are prone to water ponding, making them ideal for constructing water storage tanks. We carefully considered the suitability of these locations and their proximity to the local population. Figure 6 depicts the constructed surface water natural drainage based on a digital elevation model of Lahore city.

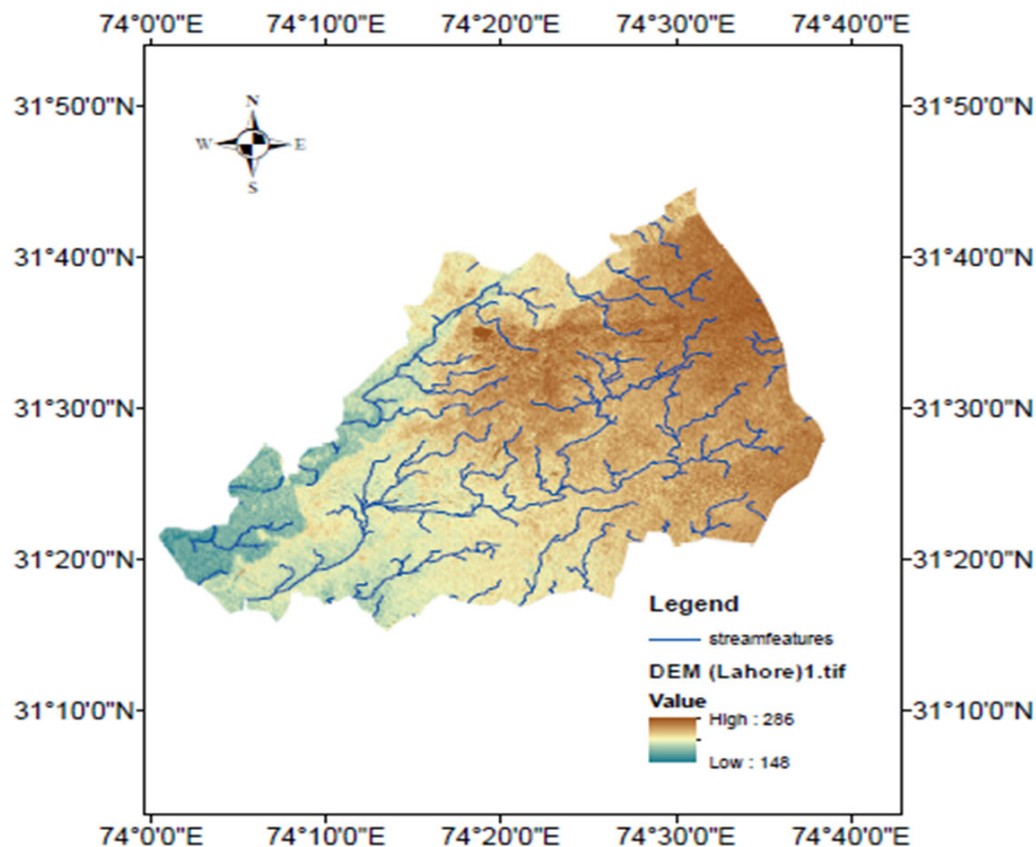

**Figure 6.** Surface water natural drainage of Lahore city based on digital elevation model.

### 2.3.3. Water Consumption Data

From publicly accessible listings of existing major public institutions, public institutions in Lahore were found and divided into 12 categories. Water consumption information for each of the digitized institutions was gathered from the literature (Supplementary Materials) and is summarized in Table 1 to explore the possible impact at the level of individual public institutions.

**Table 1.** Water consumption data for Lahore based on water usage bills from the WASA literature [54–61].

| Large Public Institutions | Consumption (m³) |
| --- | --- |
| Major industries | 292,708 |
| Large mosques/shrines | 1992 |
| Schools/colleges | 53,630 |
| Stadiums | 122,341 |
| Malls | 119,920 |
| Hotels (five stars) | 22,068 |
| Clubs | 3740 |
| Cinemas | 6120 |
| Airports | 9845 |
| Hospitals | 18,987 |
| Universities | 10,740 |
| Judicial courts | 4692 |

### 2.4. Rooftop Rainwater Potential and Identification of Ponding Sites

For calculating rooftop rainwater potential, Equation (1) was used, which consist of three parameters: (*i*) area, (*ii*) monthly rainfall (**R**), and (*iii*) the surface runoff coefficient (**Cr**).

$$Volume = R \times Area \times Cr \tag{1}$$

The monthly rainfall data for Lahore is shown in Figure 5. For the surface coefficient, different values for different materials can be used, but in the case of building rooftops, concrete is mostly used, which has values of 0.6–0.8, as shown in Table 2.

**Table 2.** Surface runoff coefficients for different pavements.

| Materials | Surface Runoff Coefficient |
|---|---|
| Concrete | 0.6–0.8 |
| Paving | 0.5–0.6 |
| PVC geomembrane | 0.85–0.9 |
| Roof tiles | 0.8–0.9 |
| Corrugated metal | 0.7–0.9 |

The rooftops of 25–30% of the institutions were first located in ArcGIS, and the data were then combined with an online Google base map from 2022 to estimate the average rooftop size (version 10.8) The rooftops of these buildings were then digitized using ArcGIS, and their areas were determined using the ArcGIS Calculate Geometry tool according to the Environmental Systems Research Institute (ESRI) standard procedures. It should be mentioned that many institutes consist of several buildings that were digitized separately but considered as single institutes. Lahore University of Management Sciences (LUMS) is one such institution that comprises several buildings, including faculties, dorms, administration, and more. In total, 213 rooftops were digitized, which represented 138 institutions. The digitized average areas of these institutions were calculated and used to establish the average rooftop area (Table 3). The digitized public institutions are shown in Figure 7.

**Table 3.** Digitized rooftop areas for the 12 categories selected in the current study in $m^2$.

| Categories | | | | Rooftop Areas ($m^2$) | |
|---|---|---|---|---|---|
| Sr. No | Large Public Institution Categories | No. of Digitized Buildings | Average Area of Digitized Buildings | Total Existing | Total Estimated Area |
| 1 | Major industries | 28 | 8175 | 300 | 2,452,500 |
| 2 | Large mosques/shrines | 10 | 3271 | 24 | 78,504 |
| 3 | Schools/colleges | 42 | 4535 | 500 | 2,267,500 |
| 4 | Stadiums | 4 | 12,067 | 19 | 229,273 |
| 5 | Malls | 3 | 26,780 | 10 | 267,800 |
| 6 | Hotels (five stars) | 4 | 6668 | 18 | 120,024 |
| 7 | Clubs | 3 | 11,340 | 11 | 124,740 |
| 8 | Cinemas | 7 | 1476 | 15 | 22,140 |
| 9 | Airports | 1 | 30,381 | 1 | 30,381 |
| 10 | Hospitals | 26 | 3606 | 47 | 169,482 |
| 11 | Universities | 7 | 20,262 | 28 | 567,336 |
| 12 | Judicial courts | 3 | 7344 | 23 | 168,912 |
| | Total | 138 | | 996 | 6,498,592 |

The identification of locations with water ponds due to depressions in the ground, as undertaken through the analysis of the stream network in our digital elevation model, holds significant importance in the field of water resource management. The knowledge of

these locations can be utilized in constructing underground storage tanks that can serve as an effective solution for the storage of water. Additionally, recharge wells can be built in such locations to help replenish the groundwater levels in the region, which is crucial in areas where water is scarce.

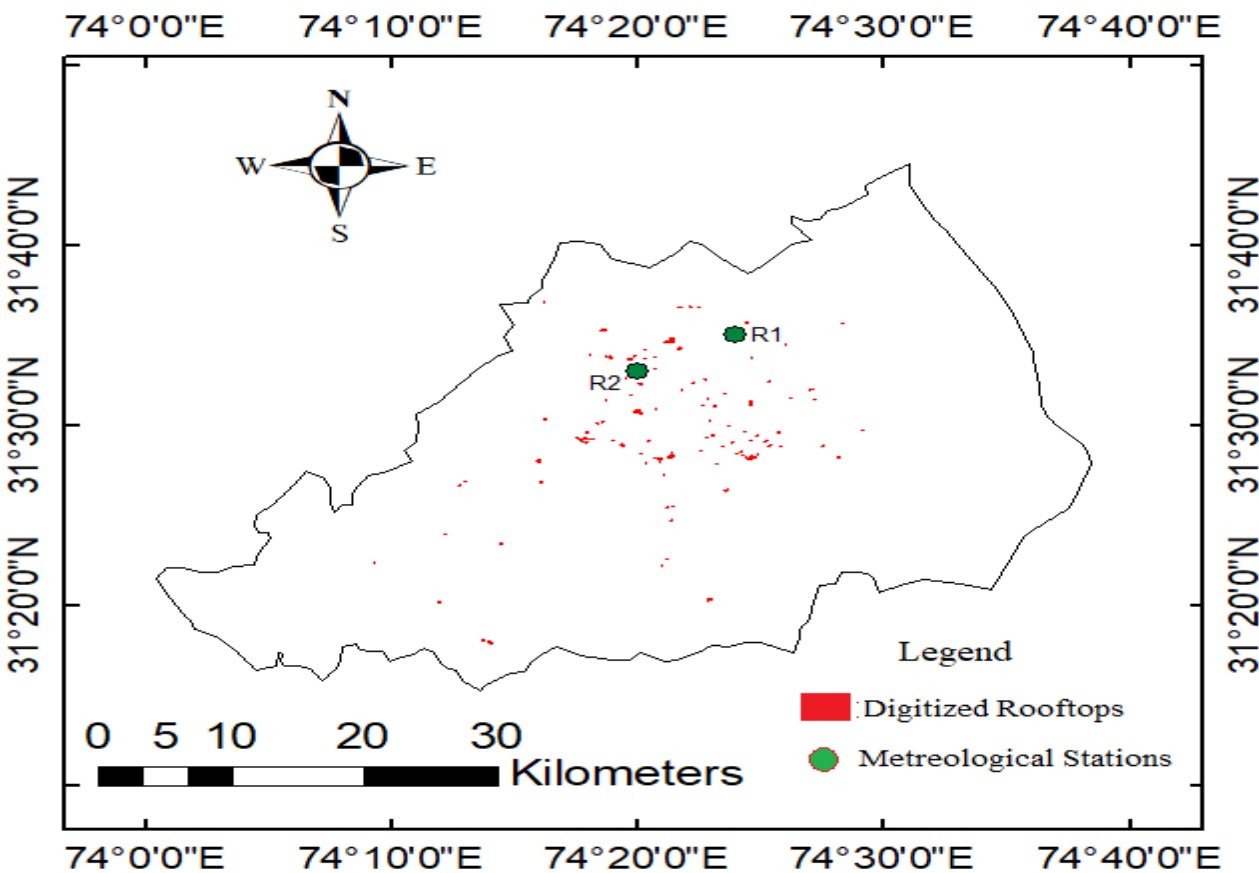

**Figure 7.** Digitized rooftops in Lahore city.

## 3. Results

### 3.1. Potential Volume of Monthly Rooftop RWH

The mean monthly potential rooftop RWH from each category of large public institutions was calculated by multiplying rooftop areas (Table 3) by the mean monthly rainfall (Figure 5), and a runoff coefficient ($Cr$) of 0.7 was employed using Equation (1), resulting in the values shown in Table 4.

### 3.2. Efficiency from Consumption and Supply

The process of calculating the percentage efficiency of RWH involved comparing the amount of water consumed from the mains supply with the amount of water saved through the collection of rainwater. This calculation was made possible by the water consumption data provided in Table 1 and the water supply data presented in Table 4, resulting in the values shown in Table 5. Once the percentage efficiency of RWH was determined, we gained valuable insight into the amount of water that could be conserved through this method. The results of this analysis may be crucial for developing sustainable water management strategies and reducing reliance on mains water supply. By using harvested rainwater, not only do we reduce the demand for mains water but we also reduce the amount of water that is wasted through runoff and flooding. This conservation of water resources can have a significant impact on the environment and help to mitigate the effects of water scarcity in regions where it is a major concern.

**Table 4.** Monthly flow for all 12 categories in m$^3$.

| Categories | January | February | March | April | May | June | July | August | September | October | November | December |
|---|---|---|---|---|---|---|---|---|---|---|---|---|
| 1 | 36,481 | 43,777 | 58,370 | 29,185 | 29,185 | 51,073 | 291,848 | 240,774 | 87,554 | 17,511 | 5837 | 20,429 |
| 2 | 1168 | 1401 | 1868 | 934 | 934 | 1635 | 9342 | 7707 | 2803 | 561 | 187 | 654 |
| 3 | 33,729 | 40,475 | 53,967 | 26,983 | 26,983 | 47,221 | 269,833 | 222,612 | 80,950 | 16,190 | 5397 | 18,888 |
| 4 | 3410 | 4093 | 5457 | 2728 | 2728 | 4775 | 27,283 | 22,509 | 8185 | 1637 | 546 | 1910 |
| 5 | 3984 | 4780 | 6374 | 3187 | 3187 | 5577 | 31,868 | 26,291 | 9560 | 1912 | 637 | 2231 |
| 6 | 1785 | 2142 | 2857 | 1428 | 1428 | 2499 | 14,283 | 11,783 | 4285 | 857 | 286 | 1000 |
| 7 | 1856 | 2227 | 2969 | 1484 | 1484 | 2598 | 14,844 | 12,246 | 4453 | 891 | 297 | 1039 |
| 8 | 329 | 395 | 527 | 263 | 263 | 461 | 2635 | 2174 | 790 | 158 | 53 | 184 |
| 9 | 452 | 542 | 723 | 362 | 362 | 633 | 3615 | 2983 | 1085 | 217 | 72 | 253 |
| 10 | 2521 | 3025 | 4034 | 2017 | 2017 | 3529 | 20,168 | 16,639 | 6051 | 1210 | 403 | 1412 |
| 11 | 8439 | 10,127 | 13,503 | 6751 | 6751 | 11,815 | 67,513 | 55,698 | 20,254 | 4051 | 1350 | 4726 |
| 12 | 2513 | 3015 | 4020 | 2010 | 2010 | 3518 | 20,101 | 16,583 | 6030 | 1206 | 402 | 1407 |
| Total | 96,667 | 116,000 | 154,666 | 77,333 | 77,333 | 135,333 | 773,332 | 637,999 | 232,000 | 46,400 | 15,467 | 54,133 |

**Table 5.** Calculating efficiency from consumption and supply.

| Large Public Institutions | Consumption (m³) | Supply (m³) | % of Water Saved |
|---|---|---|---|
| Major industries | 292,708 | 76,002 | 26 |
| Large mosques/shrines | 1992 | 2432 | 100 |
| Schools/colleges | 53,630 | 70,269 | 100 |
| Stadiums | 122,341 | 7105 | 5.8 |
| Malls | 119,920 | 8299 | 6.9 |
| Hotels (five stars) | 22,068 | 3719 | 16.8 |
| Clubs | 3740 | 3865 | 100 |
| Cinemas | 6120 | 686 | 11.2 |
| Airports | 9845 | 941 | 9.5 |
| Hospitals | 18,987 | 5252 | 27.6 |
| Universities | 10,740 | 17,581 | 100 |
| Judicial courts | 4692 | 5234 | 100 |

*3.3. Location of Underground Storage Tanks/Recharge Wells*

The construction of underground storage tanks and recharge wells can play a vital role in augmenting the overall water supply in a region, especially during periods of drought or water scarcity. The water stored in these tanks can be utilized for various purposes, such as irrigation, domestic uses, and industrial needs. The suitable locations were identified and are shown in Figures 8 and 9.

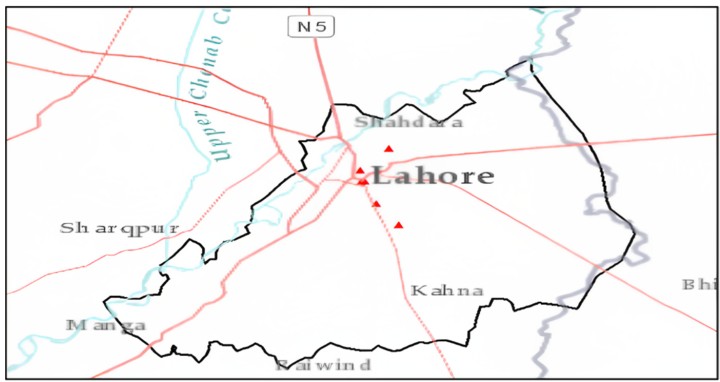

**Figure 8.** Identification of ponding sites.

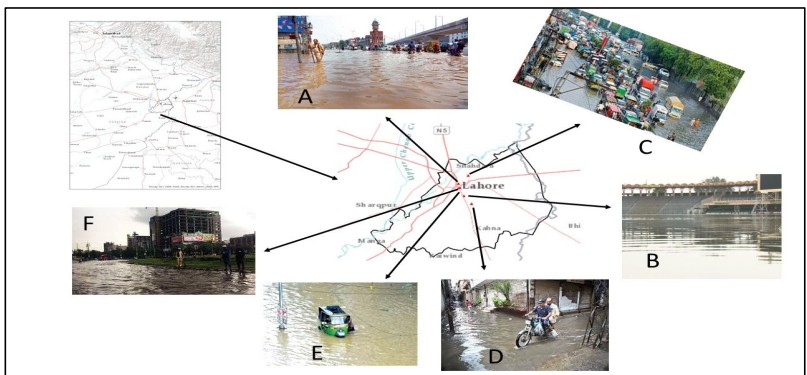

**Figure 9.** Ponding sites in Lahore. (**A**) 31.5661° N, 74.3141° E; (**B**) 31.5137° N, 74.3333° E; (**C**) 31.532° N, 74.234° E; (**D**) 31.5342° N, 74.2314° E; (**E**) 31.2314° N, 74.3542° E; (**F**) 31.5771° N, 74.3242° E.

### 3.4. A Recommended RWH Model for Institutions and Households

A model is a tool that can serve as a guide or representation of a real-life scenario, providing a structure for problem solving. To collect rainwater for household use, a rainwater harvesting system is necessary, particularly in urban areas such as Lahore. The model for the rainwater collection system at the household level involved a catchment area of 300 m$^2$, which is the typical size of a roof. The model displays the roof's ridge and valley lines, which indicate the highest and lowest points of the roof, respectively, and show how rainwater is directed into gutter sewerage systems through 4-inch-diameter PVC pipes. The rainwater is collected in a water tank, which is connected to the green pipes of the central water supply system. Both RWH and a water supply system can be used simultaneously. The collected rainwater is filtered and stored in a tank with a capacity of more than 10,000 L, determined based on the average yearly rainfall and the catchment area. The water is pumped from the tank and can be used for various purposes, such as irrigation and washing clothes, but not for drinking. To minimize the impact on the garden area, storage tanks can be buried, and warning signs should be placed on each tap to inform users that the water is not fit for drinking. Care should be taken when implementing a rainwater harvesting system at the household level. A general model of a residential building is shown in Figure 10, depicting how rainfall is collected and transported through pipes for use. Similarly, Figure 11 shows a typical model for a house at the individual level for RWH.

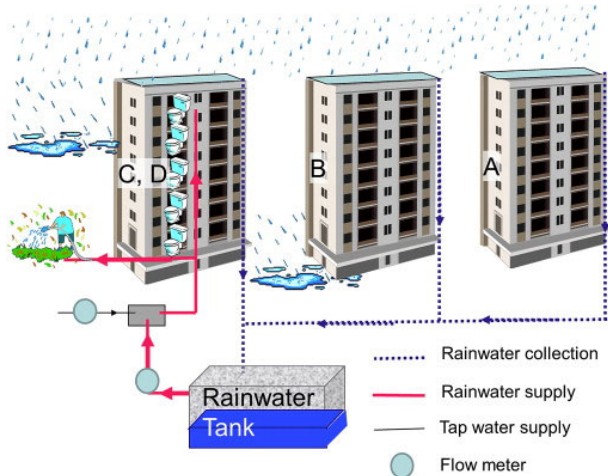

**Figure 10.** Schematic diagram of the structure of the system [62].

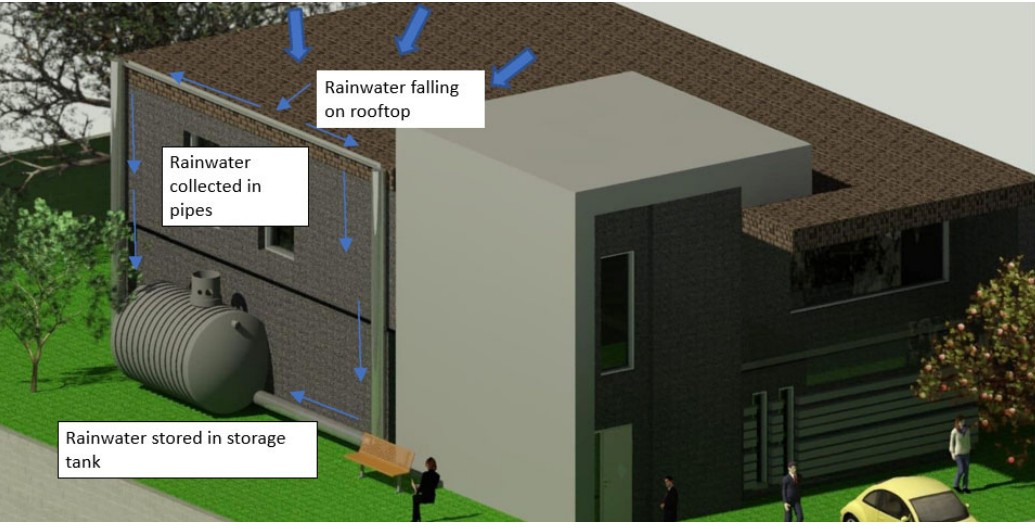

**Figure 11.** Model of a house with the rainwater harvesting technique.

*3.5. Treatment of Stormwater*

3.5.1. Mechanical Pre-Tank Filters

The easiest way to keep rainwater clean is to never allow dirt or debris to enter the storage tank. To do this, most systems use a pre- or in-tank filter. While some inferior models are wholly plastic, they frequently have a stainless-steel component contained in a plastic body. The element is typically a stainless-steel grill or mesh. The filtration is measured in microns, where a lower number indicates a finer mesh [63].

3.5.2. Microscopic Filtration

These filters are frequently in the form of sealed containers housing cartridges or bags that capture incredibly tiny particles, making it possible to achieve increasingly cleaner water standards in the present day. Even though some of the coarser grades may be cleaned and reused, when they start to clog, the cartridge or bag on most of them has to be changed periodically. Such filters have a vital feature in that they require pressure to function correctly; water must be forced through them for them to work. As a result, these filters are often exclusively utilized with pumped water supplies [64].

3.5.3. Disinfection

Disinfection techniques include chlorination, ozonation, ultraviolet (UV) light, and membrane filtering. It should be noted when evaluating them that certain disinfection treatments create dangerous remnants that need to be treated [65].

3.5.4. Ultraviolet Light

Mechanical filters can get rid of water impurities, but they cannot get rid of microbes. This is not a problem if the water will only be used for housekeeping, feeding the yard, and flushing the toilet. Chemical treatment of the water is necessary if it will also be ingested or used for bathing, etc. The water is sterilized as it passes over a glass tube and subjected to ultraviolet light of a certain wavelength and magnitude during the UV disinfection. This procedure immediately sterilizes the water by eliminating all bacteria [66].

3.5.5. Chlorination

Chlorine is used in public water systems to cleanse the water. When chlorine is used to treat wastewater, diseases such as cholera, typhoid, dysentery, and hepatitis can be practically eliminated, saving countless millions of lives. However, it is widely criticized due to suspected adverse consequences [67].

3.5.6. Distillation

A common method of purification is distillation. Distillation involves removing impurities from water by boiling it and then collecting the condensation. Evaporation results in water retention of between 5 and 10% and requires a lot of energy. Except for turbulent organic chemicals (VOCs), which evaporate quickly, almost all constituents can be removed from water during distillation. This is why some distillation systems also utilize carbon filters to remove the VOCs [68].

3.5.7. Ozonation

Ozonation sanitizes water by adding ozone gas to it. It is often undertaken at the point in the distribution system where water is consumed. It is a colorless gas with antiseptic, oxidizing, deodorizing, and decolorizing properties [69].

**4. Discussion**

In order to compute the area of practically all the rooftops of the buildings in Lahore, thousands of polygons were digitized with the help of Google Earth. The results varied depending on the size of each rooftop catchment area; however, it is possible for each rooftop studied to gather tens of thousands of cubic meters of rainfall annually. The

rainfall data were collected from the nearest meteorological stations. This quantity of water can alleviate the burden on the central water supply system and recharge groundwater. Moreover, it can reduce the cost of pumping water. The technique of gathering and preserving rainwater from a building's roof for later use is known as rooftop rainwater harvesting. Due to the rising awareness of the need to conserve water and the price of water supplies, this method of water conservation, although it has been in use for millennia, has recently gained popularity. There are many methods for calculating the volume of rainwater, but our technique is the most efficient, economic, and practical. It can be applied to any city, not only in Pakistan but all over the world. Water collected from these rooftops flows following the stream network of Lahore, which was generated by using ArcGIS software. By carefully analyzing the stream network, the specific locations of the ponding sites were located in Lahore. Rooftop rainwater harvesting involves installing a system that gathers rainwater as it falls on a building's roof. Following filtering, the collected water is put into a tank or reservoir for later use. The water may be used for several purposes, including drinking, cooking, washing automobiles and clothing, watering plants, and flushing toilets.

Rainwater collection on rooftops has several advantages. By lowering the demand on municipal water resources, it first and foremost aids in water conservation. This is especially crucial in regions with limited water supplies or where droughts often occur. Homeowners and businesses may lower their water bills and contribute to ensuring that there is adequate water for everyone by collecting rainwater. Due to this contribution, less water will be required as compared to previously; furthermore, less groundwater pumping will be undertaken, due to which electricity will also be saved because pumping is a heavy duty, and the motors require huge amounts of electricity, so rainwater collection will also help in saving electricity.

One of the major issues cities faces is urban flooding, which affects cities across every aspect of life.

In addition to being an efficient alternative water supply option in arid and semi-arid locations, rainwater harvesting (RWH) systems offer various advantages. These technologies can also help to lower the danger of flooding in urban areas. Nevertheless, while the majority of studies have emphasized the potential of RWH for lowering water use, few have tested its effectiveness in retaining rainwater in residential areas vulnerable to flooding. Investigating RWH systems' dependability in terms of stormwater retention was the goal of this study. Various studies have indicated that the widespread installation of RWH tanks might provide effective assistance in lowering the frequency and peaks of stormwater floods in urban catchments [70]. An urban community can face various issues, such as power outages, traffic congestion, disruption of business, and contaminated water, due to certain phenomena. Health risks can also arise due to the speed of the water flow and potential pollutants, as well as the danger of electrocution from live wires submerged in flood water. To mitigate these problems, collecting rainwater from rooftops can be a useful practice to provide water for a variety of purposes, such as watering plants, washing clothes, flushing toilets, and even drinking. However, in order to make sure that the rainwater is safe to use, it must first be filtered and treated. For the purpose of collecting rainwater from rooftops, a variety of filtering processes can be applied, including physical, biological, and chemical processes, which were discussed above.

It is also important to keep in mind that several variables, such as the caliber of the entering rainwater, the types of pollutants present, and the planned use of the collected rainwater, can affect how efficient the filtering system is. As a result, it is crucial to take these things into account when choosing a filter system for rooftop rainwater collection.

Finally, rooftop rainwater harvesting is an environmentally friendly and economical option to store and use rainwater, but it is crucial to filter and treat the water to assure its safety. The choice of a filtration system should be based on the needs and demands of the rainwater harvesting system because different filtering techniques each have advantages and disadvantages. The collected rainwater should be treated, but the level of treatment

will differ depending on the purpose of the use. If it is used for drinking, it should be treated 99.9%, and if it is used for flushing and industrial use, it should be treated as such.

The promotion of sustainable development and the accomplishment of the various Sustainable Development Goals (SDGs) of the United Nations depend on the practice of collecting rainwater. A quick and efficient method for collecting rainwater for use later on is called "rainwater harvesting". This method offers a number of advantages, including lowering runoff and erosion, improving sustainable agriculture, and lowering the demand for groundwater. Rainwater collection is a key strategy for advancing SDG 6: Clean Water and Sanitation. By collecting rainwater, communities may lessen their reliance on groundwater, which can be polluted or exhausted from overuse. This makes it possible to guarantee that everyone has access to clean, safe drinking water, which is necessary for maintaining human health and wellbeing. For SDG 2: Zero Hunger, rainwater collection can help farmers increase their yields and promote food security by giving crops a dependable source of water. Communities may also profit economically from this since farmers may make money by selling their extra product. SDG 11: Sustainable Cities and Communities may also be facilitated by rainwater collection. Rainwater collection may lessen erosion and runoff, which helps to minimize flooding and enhances the quality of urban areas. It is also possible to lessen the demand for freshwater resources by utilizing rainwater for irrigation and other non-potable uses. This is crucial in metropolitan areas where water is frequently in short supply. SDG 13: Climate Action may be addressed with the use of rainwater harvesting. Rainwater collection can assist in lessening the effects of climate change and save ecosystems by lowering runoff and erosion. The need for energy-intensive treatment and distribution of freshwater resources can be decreased by using rainwater for non-potable applications, which can aid in lowering greenhouse gas emissions.

In general, rainwater collection is a useful technique for advancing sustainable development and accomplishing several of the Sustainable Development Goals set forth by the United Nations. Communities can increase their access to clean and safe water, support sustainable agriculture, reduce their influence on the environment, and work towards a more sustainable future for everybody by collecting and using rainwater. Rainwater harvesting systems have been proven to reduce reliance on carbon-producing mains water supplies. For every 1000 $m^3$ of rainwater used, there is a saving of 285 kg of $CO_2$ [71].

The RWH described in this research can help mitigate the shortage of water during dry seasons, reduce dependence on traditional water supply systems, and restore depleted aquifers through direct usage or groundwater recharge. It can also aid in the management of urban floods by catching rainfall, lowering flood hazards, and relieving pressure on ageing stormwater infrastructure. Furthermore, by offering an alternate water source for non-potable purposes, it reduces demand on the municipal water supply, encourages sustainable water management practices, and results in cost savings for people, institutions, and cities. Implementing these practices in Lahore can help to increase water security, enhance water infrastructure, manage stormwater more efficiently and contribute to a more resilient and sustainable future.

## 5. Conclusions

To investigate the potential contribution of rooftop rainwater harvesting in Lahore, the public institutions were divided into 12 categories and these rooftops were digitized in ArcGIS to obtain the rooftop area. In order to quantify the potential water that could be harvested from each category, the rainfall data were collected from the nearest meteorological stations in Lahore, from which we obtained the monthly amount of runoff from each category. Water consumption data were obtained for each category from the WASA and compared with the RWH potential to determine the efficiency of RWH.

In conclusion, rainwater harvesting is a practical and long-lasting way to deal with urban flooding and a lack of drinking water. Rainwater collection and storage allow us to maintain a consistent source of clean water, especially in places with limited access to

clean drinking water. This reduces the pressure on municipal water supplies. Furthermore, by reducing the quantity of stormwater runoff that enters the streets and contributes to floods, rainwater harvesting can also aid in reducing urban flooding. We can stop groundwater supplies from running out by collecting rainfall and rerouting it into storage tanks. It is crucial to remember that, before using collected rainwater for drinking and other domestic uses, it must be tested for purity. Various filtering methods, including sand filtration, charcoal filtration, and UV treatment, can be used to achieve this. To give everyone convenient access to clean water, filtered water stations could be built in public areas.

Rainwater harvesting, in conjunction with appropriate filtration methods and filtered water points, can act as an efficient and long-term solution to deal with urban flooding and the lack of potable water, while also encouraging water conservation and minimizing the effects of climate change on our communities.

In future studies, to examine how storage tanks and recharging wells behave under various circumstances, researchers could undertake lab studies, field tests, and computer simulations. In order to assess their effectiveness and pinpoint areas for development, they could also look at the RWH systems that are already in place in various locations.

The findings of this study can assist the design and building of RWH systems that are more effective and efficient, addressing water scarcity and promoting sustainable water usage in communities all over the world.

**Supplementary Materials:** The following supporting information can be downloaded at: https://www.mdpi.com/article/10.3390/civileng4020037/s1, File S1: software ArcGIS related files.

**Author Contributions:** M.A., N.A. and S.M.U.G. conceived and designed the overall study concept. M.W., N.A. and M.A. conducted the model setup, simulation, and calibration. M.K.L. collected river cross sections and river flow data. M.A., N.A. and S.M.U.G. were responsible for the consultations regarding model simulation results, comparisons, manuscript writing, proof reading, and manuscript modification. M.K.L. and M.W. supervised the overall research. All authors have read and agreed to the published version of the manuscript.

**Funding:** This research received no external funding.

**Data Availability Statement:** The data analyzed in this paper are available from the first author upon reasonable request.

**Acknowledgments:** The authors would like to thank the Water and Sanitation Agency (WASA) and Pakistan Meteorological Department for providing the necessary support for this research.

**Conflicts of Interest:** The authors declare no conflict of interests.

## Appendix A

**Table A1.** Sources of data.

| Sr No. | Type of Data | Description | Source |
|:---:|:---:|:---:|:---:|
| 1 | DEM | SRTM DEM GL1 with spatial resolution of 30 m | Open Topography [72] |
| 2 | Rainfall data | Nearest meteorological stations | Pakistan meteorological department [73] |
| 3 | Water consumption | Water consumed in each category | Water and Sanitation Agency of Lahore [12] |

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
