# Peer review of "Rainwater Harvesting as Sustainable Solution to Cope with Drinking Water Scarcity and Urban Flooding: A Case Study of Public Institutions in Lahore, Pakistan"

_2673-4109, doi:10.3390/civileng4020037_

Round 1

Reviewer 1 Report

 The article "Rainwater Harvesting as Sustainable Solution to Cope with Drinking Water Scarcity and Urban Flooding: A Case Study of Public Institutions in Lahore, Pakistan" discusses the use of rainwater harvesting (RWH) as a potential solution to address drinking water scarcity and urban flooding issues in Lahore, Pakistan. The article highlights that Pakistan is currently facing physical and economic water scarcity challenges exacerbated by population growth and climate change, and RWH could be a viable alternative source of domestic water supply.By emphasizing the importance of sustainable water management and considers RWH as an integral part of it, citing examples from around the world.  Overal, it provides a brief overview of rainwater harvesting (RWH) as a potential solution to address water scarcity and urban flooding in Lahore, Pakistan. However, there are several critical points that can be highlighted in the review:

- Lack of discussion on challenges and limitations: The article does not discuss the potential challenges or limitations of implementing RWH in Lahore or other similar contexts. For example, it does not address issues such as water quality, maintenance and management of RWH systems, social and cultural acceptability, and potential conflicts over water use, which can affect the effectiveness and sustainability of RWH implementation.

- Generalized statements: The article makes generalized statements such as "rainwater has great potential to be taken as a source of water demands in residential colonies of major cities like Lahore," without providing specific evidence or data to support the claim. This reduces the reliability and credibility of the article.

- Lack of references: The article lacks proper references to support the information and claims presented. It would be beneficial to provide credible and up-to-date references to support the statements made in the article and to allow readers to access further information on the topic.

- Limited discussion on urban flooding: The article mentions urban flooding as one of the issues in Lahore, but does not provide adequate discussion on how RWH can effectively address urban flooding challenges. Urban flooding is a complex issue influenced by various factors, and the article could benefit from a more comprehensive analysis of how RWH can contribute to mitigating urban flooding in Lahore.

-Contribution to academia needs to be highlighted in the abstract, introduction and conclusion part of the study. The contribution of the study needs to be explained in such a way that to increase the originality of the study.

- Abstract should cover Introduction and Reason for conducting the research, the Problem (knowledge gap), Methods, Outcomes (results), and Ramifications (Implications). The abstract should be re-written so that it encompasses summaries of the most important parts of the study results and authors' arguments.

- The authors may need to justify why this case study (s) and how the findings can be generalizable.

- The ‘discussion part also needs to develop considering the aim of the article and how the author responds to the hypothesis of the manuscript. The functionality of the methodology and tactics used in the article needs to be discussed. the discussion needs to be written in a scientific way not as separate paragraphs. coherency should be there.

- The conclusion needs to restructure to have its own independent title. some essential information which supposes to be in the conclusion part is missing. For example, what are the findings to support the hypothesis of the study? how the author(s) described the contribution of their study to the existing literature? etc., the Conclusion of the study could be much more descriptive in the findings that the author (s) mentioned on the discussion part.

- All the cited references should be directly relevant to the research.

-Suggestion for future study is also missing from the last line of the conclusion. It should be used to point out any important shortcomings of the manuscript, which could be addressed by further research or to indicate directions for further work could take.

Reviewer 2 Report

Rainwater Harvesting as Sustainable Solution to Cope with Drinking Water Scarcity and Urban  Flooding: A Case Study of Public Institutions in Lahore, Pakistan

1.       The abstract just only provides a background of the water scarcity in Pakistan, but it failed to provide any information on what research has been done, how it was done, what were the results and finding. The abstract needs a complete revision to reflect the above information.

2.       What is the source of figure 1. Figure 2 also needs to be explained the text.

3.       Study area (Lahore) statistics need to be supported with relevant references/ citations. You have mentioned the 2017 populations which is not current, thus try to use current data/ population. Just a simple google search and using the world population review indicate the Labore population as 14 million.

4.       Quoting WASA planning (master plan) for the next 25 years is ill conceived as we know this organization is badly failed to solve the problem of water and sanitation in Lahore. Most of the part of the city remained flooded in normal weather due to the blockage of drain and sever line. Most of the water supply line are mixed with sewer lines contaminating the clean drinking water. This has resulted several health issues among the population of Lahore. The situation of the city in rainy days is more worse. This is also reflected in your figure 4.

5.       Why you are repeating the population of Lahore again and again (see section 2.1 and 2.2).

6.       What rain stations you have used. How you access to the data.

7.       How you did stream networks. Did you used ArcGIS, if so, state this clearly.

8.       Why you have only considered some specific buildings for water consumptions – any rationale behind this. What about the domestic use.

9.       There is no clarity on the data. It is no clear what industrial buildings and other buildings were used. For any building used in the data should be clearly stated. Figure 7 is not clear.  

10.   Check the data in table 3. Is it in m3? Data in table 5 is also not clear. not sure how you arrived on consumption and supply. Figures 8 and 9 are not clear.

11.   I do not see any specific contribution of this paper. there is a discussion section which provides some general discussion but there is no conclusion. Make sure you write the conclusion section in which you highlight the key contributions of this paper and state the limitations of your research. It was good to see the SDGs but the discussion is quite limited and link is not well established with your research. I suggest to review this paper “Exploring the GCC progress towards United Nations sustainable development goals” which might help you to develop such link.

12.   All the data set need to be appended in the appendix so that the accuracy of the figures/data reported in the paper can be traced. Also make clear what data has been obtained from WASA and Pakistan Meteorological 368 Department.

13.   The authors contributions should be stated clearly. I would be particularly interested to know the contribution of Megersa Kebede Leta in this research to make sure there is no ethical issues.

14.   The paper need a through proofread to eliminate the grammatical error and improve the clarity.

Round 2

Reviewer 1 Report

The comments provided have been taken into consideration and the manuscript has undergone significant improvements, particularly in its theoretical and methodological aspects. As a result, the article now has a clear contribution and an enhanced internal validity. Based on these developments, I believe that the article is now suitable for publication.

Author Response

Dear Reviewer,

Thanks a lot for providing helpful suggestions to improve this paper. All the comments have been addressed and changes have been made accordingly in the text. 

Reviewer 2 Report

The authors have tried to revise the paper and attempted to take my previous comments on board, however, the changes made to the paper do not warrant publication. There is still concern related to research approach, data collection, data authenticity, and the overall contribution of the paper is still there, thus the paper cannot be considered for publication in current form.

A thorough proofread is required. The written english do not provide clarity in many places. 

Author Response

Dear Reviewer,

Thanks a lot for providing helpful suggestions to improve this paper. All the comments have been addressed and changes have been made accordingly in the text. The followings are the explanations and actions we have taken regarding your valuable comments.

General Explanations/Corrections:

  • The manuscript has been revised by a native speaker and thorough corrections have been made regarding the sentence structure and grammatical mistakes.
  • All manuscript sections have been improved.
  • All tables have been improved as per MDPI criteria.
  • Reference numbering in the text has been corrected.

Specific Comments and Explanations

Comment # 1.

Concern regarding the data collection and data authenticity.

Explanation: All the shape files containing digitized roof surfaces for all selected 12 categories, constructed natural drainage, and climate data have been provided even at the time of the first revision to the editor. Data can be checked and verified.

Comment # 2.

Concern regarding the overall contribution of the paper.

Explanation: The text has been revised and a paragraph has been added in the discussion section.

Comment # 3.

Concern regarding the manuscript language quality.

Explanation: The manuscript has been revised by a native English speaker and detailed revisions have been made.

Round 3

Reviewer 2 Report

Unfortunately, the paper failed to address the issues highlighted in the previous comments. There is a lack of novelty, the research approach is faulty, and the data collection method do not capture the data required to deliver the aims/ objectives of the paper. The data analysis superficial. The results/ conclusion is overstated.

Having the above serious flaws, the paper cannot be recommended for acceptance.

thorough proofread is recommended